



# Investigating limnological processes and modern sedimentation at Lake Żabińskie, northeast Poland: a decade-long multi-parameter dataset, 2012-2021

Wojciech Tylmann[1], Alicja Bonk[1], Dariusz Borowiak[2*], Paulina Głowacka[1], Kamil Nowiński[2], Joanna Piłczyńska[1], Agnieszka Szczerba[1], Maurycy Żarczyński[1]

[1]Department of Geomorphology and Quaternary Geology, Faculty of Oceanography and Geography, University of Gdańsk, 80309 Gdańsk, Poland
[2]Department of Hydrology, Faculty of Oceanography and Geography, University of Gdańsk, 80309 Gdańsk, Poland
*deceased

*Correspondence to*: Wojciech Tylmann (wojciech.tylmann@ug.edu.pl)

**Abstract.** Here, we present the dataset from a decade-long monitoring at Lake Żabińskie, a hardwater and eutrophic lake in northeast Poland. The lake contains annually laminated (varved) sediments which form a unique archive of environmental variability in the past. Regular measurements of the lake water physical and chemical characteristics were done using multiparameter sonde and a set of temperature sensors deployed in the water column. Seasonal variability of sediment fluxes was documented by a sediment trap. Field sampling provided information about the hydrochemistry of incoming streams and of the outflow from the lake. The overall monitoring program was designed to capture a pattern of relationships between meteorological conditions, limnological processes, and modern sedimentation, and to answer the question if meteorological and limnological phenomena can be precisely tracked with varves. However, this dataset can also be a solid background for modeling physical and biogeochemical processes in lakes. The dataset is archived at https://doi.org/10.34808/bsk4-eg58 (Tylmann et al., 2023).

## 1 Introduction

Lakes that contain annually laminated (varved) sediments have been recognised as invaluable environmental archives (Zolitschka et al., 2015). Using varved sediment cores and high-resolution analytical techniques, changes in sediment composition can be observed at (sub)seasonal-scale, providing a unique insight into environmental variability in the past (Butz et al., 2015; Croudace et al., 2019; Żarczyński et al., 2022). However, formation of varves in eutrophic temperate lakes is influenced by complex interactions of physical, chemical and biological processes. To accurately track past environmental changes using lake sediments, it is thus necessary to recognise the impact of different processes and environmental conditions on the dynamics of sedimentation. High-resolution instrumental data that document the relationships between meteorological conditions, limnological processes in the water column, and sediment fluxes are rare and usually cover a relatively short observational period (e.g., one or two years), which does not allow for taking into account a range of



variability in environmental and meteorological conditions. For a better interpretation of sediment records, long-term instrumental datasets that cover a wide range of measured parameters are needed.

Here, we present the results of a decade-long investigation of modern processes at Lake Żabińskie (alternative name Lake Żabinki) located in northeast Poland. The lake has been selected for long-term monitoring because it forms biogenic (calcite)

varves that are typical for hardwater, eutrophic lakes of the temperate climate zone (Tylmann et al., 2013). The systematic on-site measurements and sampling were initiated in 2011 in the frame of the project CLIMPOL (2011-2015) that aimed at quantitative climatic reconstructions over the last 1,000 years, a time period capturing climatic variability relevant for the development of modern society. In 2012, a 19.4 m-long sediment profile (10,800 yrs) was retrieved from the deepest part of this lake and investigated with a multi-proxy approach. In 2016 the monitoring was extended by an additional six years

within the project 'Tracking climate signals preserved in lake sediments from integrated process studies and ultra high-resolution analysis of annually laminated sediments'. As a result, we collected a wide range of information on the physical and chemical characteristics of the lake's water and modern sedimentation dynamics as well as the hydrochemistry of incoming streams and outflow from the lake. The overall monitoring program was designed to capture a pattern of relationships between meteorological conditions, limnological processes, and modern sedimentation and to determine

whether meteorological and limnological phenomena can be precisely tracked with varves.

The goal of this paper is to present the methods of data collection, describe the data, and to make this unique dataset available to the scientific community. The data were already used to explore the record of meteorological conditions in varves of Lake Żabińskie (Żarczyński et al., 2022) but we do believe that this multiyear dataset provides useful context for the interpretation of other lake sediment records and serves as a modern analogue of past sedimentation in lakes of temperate

climate zone. Moreover, it could be valuable for modeling physical and biogeochemical processes in lakes. Long-term monitoring datasets are of crucial importance to validate limnological models which in turn allow to assess limnological conditions in lakes where observations have not been made. They also allow for predicting long-term trends in water temperature changes under different scenarios of future climate change (Piccolroaz et al., 2018). Despite this potential, very few studies combine comprehensive monitoring of water column characteristics and sedimentation processes in lakes (e.g.,

Heinrich et al., 2018; Maier et al., 2018; Broadman et al., 2019; Brauer et al., 2019).

## 2 Site description

Lake Żabińskie (54.1318°N, 21.9836°E; 117 amsl) is located in the Land of Great Masurian Lakes, northeastern Poland (Fig. 1). The region is characterised by typical postglacial landscape, with diverse morphology, a wide diversity of glacial landforms, and a very high abundance of lakes. According to meteorological data for the period 1991–2020 (Tomczyk and

Bednorz, 2022), the continental climate features a strong seasonality with a mean air temperature of 8.0 ˚C, the lowest mean temperatures occurring in January (-2.5 ˚C), and the highest in July (18 ˚C). The ice cover on lakes in the region usually occurs between December and April (Marszelewski and Skowron, 2006), but in last two decades this period has been

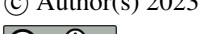



shortened. Snow cover usually lasts for about 60 days. Annual precipitation varies between 550 and 600 mm with a predominance of summer rainfalls. The wettest month is June (approx. 80 mm), while February is the driest (approx. 25

mm). Westerly and south-westerly winds dominate in the area (Hutorowicz et al., 1996). The hydrographic network in the region has the highest areal density of lakes in Poland reaching up to 20% (Choiński, 2007).

Lake Żabińskie is sited in the northern part of the region (Fig. 1). The total catchment area (24.8 km$^2$) expands eastwards from the lake and is divided into three subcatchments: Lake Łękuk (13.6 km$^2$), Lake Purwin (7.2 km$^2$), and a direct catchment of Lake Żabińskie (4.0 km$^2$). The catchment topography is diverse, showing features of moraine landscape with

elevations between 110 and 230 amsl. Valleys and lake basins are incised with steep slopes reaching locally up to 45˚. The surface geology is dominated by glacial tills and widespread fluvioglacial sands and gravels associated with outwash formations (Szumański, 2000). Modern land cover in the catchment is dominated by forests (approx. 65%). The woodland communities consist mainly of different types of pine and mixed pine forests that prevail on sandy soils, whereas oak-lime-hornbeam forests developed in morainic areas on the more fertile substrates (Forest Data Bank, http://bdl.lasy.gov.pl). The

arable lands and pastures (approx. 31%) occupy the area between Lake Żabińskie and Lake Łękuk. Human settlements are located in the direct Lake Żabińskie catchment: Żabinka village (approx. 0.5 km south to the lake) and a recreation area is on the northern shore of the lake.

Lake Żabińskie is a kettle hole lake with typical characteristics, i.e., small surface area (0.4 km$^2$) and considerable maximum depth of approx. 44 m (Fig. 1). The lake basin is slightly elongated in the W-E direction and shows two subbasins: the

central, deepest part surrounded by steep slopes and the shallow part on the west side with a maximum depth of 12 m. The basic morphometric features of the lake are presented in Table 1. The lake is exorheic and receives water from the nearby Lake Purwin and two southern creeks supplying water from cultivated fields. The southernmost inlet transports significant amounts of sediment particles and organic remains, especially after spring snowmelt and heavy rains in summer. The lake discharges westward into the larger Lake Gołdopiwo. Taking into account the hydrogeological situation, the lake is likely

supplied by groundwater from aquifers related to surrounding outwash plain sands or deeper inter-morainic sediments (Mapa Geośrodowiskowa Polski. Arkusz Giżycko, 2012).

Lake Żabińskie is a hardwater eutrophic water body with a highly productive and calcium-rich epilimnion. Anoxic conditions in deep waters have led to good preservation of biochemical varves in sediments. For this reason, the Lake Żabińskie sediment record has been investigated extensively in terms of modern sedimentation (Bonk et al., 2015;

Żarczyński et al., 2022), history of the lake mixing regime and productivity (Witak et al., 2017; Żarczyński et al., 2019; Zander et al., 2021), catchment erosion (Bonk et al., 2016), vegetation change and human impact (Wacnik et al., 2016; Hernández-Almeida et al., 2017).



## 3 Methods

### 3.1 Strategy for long-term monitoring

Limnological measurements at Lake Żabińskie were initiated in 2011, but regular observations started in 2012. The overall concept of monitoring is presented in Figure 2. The primary measurement location is established at the deepest part of the lake (Fig. 1). Two submerged buoys are anchored at this position. Lake water temperature as well as monthly sediment deposition are continuously measured by thermistors and a sediment trap, respectively. Additionally, several lake water parameters are manually measured in the water column at regular intervals. At the same time, water samples were collected

from the water column, inflows, and outflow. The population in the region and easy access to the lake for anglers and other visitors limited the possibilities for the installation of costly automatic measurement infrastructure. Thus, we decided to install only the basic equipment (thermistors and sediment trap), while other measurements required regular field trips. Long-term, daily meteorological data from the meteorological stations in Kętrzyn (approx. 40 km west of the lake) or in Mikołajki (approx. 50 km southwest of the lake) are publicly available from the Institute of Meteorology and Water

Management – National Research Institute.

### 3.2 Thermistor data

A chain of HOBO Water Temperature Pro v2 loggers (ONSET, USA) was deployed under the buoy anchored at the deepest part of the lake (Fig. 2). The loggers recorded the water temperature every 15 min at depths of 1, 10, 20, 30, and 40 m. Data were offloaded from the loggers typically once a year, in the late fall or early winter. After each offload, data were cleaned

and homogenised. Cleaning included manual removal of records obtained during the offload (e.g., the logger was not submerged and showed air temperature). Additionally, the data were screened for gross errors (temperature > 35 °C and < 0 °C, negative values were corrected to 0 °C). Afterward, the data were cleaned by removing the most extreme 1 ‰ from the series (e.g., quantiles 0.0005 and 0.9995). Finally, the time zone was unified across the records and expressed as UTC.

Due to minor shifts occurring every time the buoy was reset (respective 'series' variable in the dataset), data series from

every depth was homogenised to account for the alteration in the logger depth. For this, a ratio of mean water temperature for the last 2 hours from a preceding time series and the first 2 hours from the following sequence was calculated and used as a correction factor for the next series. This step minimised minor changes occurring between the measurement series. The homogenisation procedure was reset every time a gap was introduced (respective 'period' variable in the dataset) to account for gaps (e.g., no data recorded) in the series and to avoid artificial shifts.

Homogenised time series from each depth were used to calculate daily mean temperature values. Due to hard drive failure and corruption of files with data from the logger installed at a depth of 1 m, no raw data is available from 2017.11.04 to 2019.01.30. However, daily mean values computed during the incremental database maintenance were recovered and used to fill the daily series. Additionally, data between 2019.01.31 and 2019.03.16 is unavailable for the entire chain due to logger battery failure. Furthermore, the dataset from a depth of 40 m has more gaps, for example, due to loggers being lost during



the annual retrieval and not being replaced (2019.01.31–2020.03.07 and 2021.01.08–2021.12.31). Temperature data from the 40 m depth between 2020.03.08 and 2021.01.07 was recorded by the HOBO U26 Dissolved Oxygen logger (ONSET, USA) operating at 1 h resolution.

### 3.3 Field measurements

Water column parameters were measured between January 2012 and December 2021 with monthly (2012–2016, 2020–2021)
or bi-weekly intervals (2017–2019). All measurements were performed at the same location (Fig. 1) from a rubber boat or the ice cover during winter. Field work was suspended when the ice cover was not sufficiently thick. During 130 field campaigns, we measured water temperature (WT), dissolved oxygen concentration (DO), pH, specific conductivity (SC), and chlorophyll-a concentration (Chl-a) in the water depth range of 0–40 m with 1 m intervals. During 2012–2015, a YSI 6820 multiparameter sonde (Yellow Spring Instruments, USA) was used, while chlorophyll-a measurements were done
using Minitracka IIC fluorometer (Chelsea Instruments, UK). Afterward, an EXO2 multiparameter sonde (Yellow Spring Instruments, USA) was used for all measurements.

### 3.4 Water sample collection and analysis

Water samples were collected from 1 and 40 m water depth. Additionally, from 2013 we collected samples from major inflows (I1, I3) and outflow (O1), while sampling inflow I2 was mostly impossible due to its episodic nature. Gaps in the
dataset are related to periods of lack of water in inflows. Water samples were collected using a Van Dorn water sampler, placed in 1-L polyethylene bottles, transported to the laboratory, and stored at 4 °C before analysis. The concentration of major ions and nutrients ($Ca^{2+}$, $Mg^{2+}$, $Na^+$, $K^+$, $SO_4^{2-}$, $Cl^-$) was determined by ion chromatography (ICS 1100, Dionex, USA). Total phosphorus (TP) and total nitrogen (TN) analyses were performed after sample mineralisation using the colorimetric method and Spectroquant spectrophotometers (NOVA 400, Pharo 300, Prove 600; Merck, Germany).

### 3.5 Sediment trap sampling and analysis

Monitoring of modern sedimentation was carried out with a sediment trap made of four 1-m-long PVC liners (⌀90 mm; 0.02344 $m^2$ total active area) with removable cups at the bottom. The active area of the trap was exposed at 2 m above the sediment surface (Fig. 2). The trap was installed in May 2012 and recovered monthly (average interval ≈ 33 days, 93 samples) during the ice-free seasons. Longer periods between trap retrieval occurred under ice cover conditions (62 to 153
days, eight samples). The samples were transferred from the trap into plastic containers, transported to the laboratory and stored at 4 °C before analysis. First, samples were freeze-dried and weighed to estimate dry sediment mass. Daily fluxes, i.e., mass accumulation rates (MAR, g $m^{-2}$ $day^{-1}$), were then calculated by dividing the sample mass (g) by the trapping time (days) multiplied by the active area ($m^2$).

Concentrations of total carbon (TC), total nitrogen (TN), and total sulfur (TS) in the sediment samples were determined with
a Vario EL Cube elemental analyser (Elementar, Germany) according to standard procedure (Żarczyński et al., 2019).

Analyses of total inorganic carbon (TIC) were performed using a SoliTIC module (Elementar, Germany) coupled to the Vario EL Cube. Total organic carbon (TOC) was calculated as the difference between TC and TIC. The precision and accuracy of elemental analyses were tested on certified standard materials B2176 (CNS) and B2188 (TIC) supplied by Elemental Microanalysis. Precision ranges are: TC 0.05 %–1.93 %, TIC 0.04 %–0.45 %, TN 0.01 %–0.22 %, and TS 0.01 %–0.06 %.

### 3.6 Ice cover data

Dates of the ice cover formation and breakup were based on field observations and information obtained from the local citizens (e.g., employees of the tourist resort located north of the lake). Additionally, available online Landsat and Sentinel satellite imagery datasets were manually screened for ice cover traces. However, in some years, it was challenging to establish the number of days with frozen lake surface because of variable meteorological conditions with air temperature values around 0 °C, resulting in discontinuous ice cover.

### 4 Data description

#### 4.1 Water column data time series

Water temperature changes observed in the thermistor data series showed strong seasonality in surface layer (Fig. 3). The range of mean daily values varied between 0.3–27.2 °C. Values and variability of water temperature in deeper waters were progressively lower, and below the depth of 20 m stabilised close to 4.0 °C.

The lake is generally dimictic and develops a strong summer stratification lasting from May/June to October (Fig. 4). The water column can be completely mixed in spring (April/May) and fall (November/December). Severe winter conditions led to the development of ice cover and reverse water column stratification, mostly from January to March. However, a detailed analysis of the major physicochemical properties of Lake Żabińskie showed more a complex stratification regime depending on the seasonal meteorological conditions (Żarczyński et al., 2022). The intensity of spring/fall mixing is well illustrated by the vertical distribution of oxygen concentrations in the water column (Fig. 4). During sufficiently long mixing periods, oxygen was transported to the lake bottom (e.g., spring 2012, fall 2013, spring 2014, fall 2015, and winter/spring 2020), while rapid development of summer stratification or ice cover resulted in oxygen only reaching a certain depth (e.g., during the period 2016–2018). During the summer stratification, anaerobic conditions developed throughout the hypolimnion, and the zone of hypoxia extended from a water depth of 4 m. We observed also the influence of increasingly mild winters, e.g., in 2019/2020 when the ice cover did not develop. This resulted in long and intensive mixing and high oxygen content in the whole water column. A similar situation occurred in the winter of 2020/2021, when the ice cover was thin and discontinuous. Data on water pH, specific conductivity, and chlorophyll-a show characteristic variability related to biogeochemical processes occurring in the epilimnion during spring and summer (Fig. 4). Rapid warming of the epilimnion during spring limited $CO_2$ solubility and led to a quick increase of water pH. At the same time, an increase in water temperature triggered



intensive phytoplankton blooms expressed as peaks in the content of chlorophyll-a. Finally, these processes led to rapid calcite precipitation and a subsequent decrease in specific conductivity. This phenomenon was observed every year throughout the monitoring period.

## 4.2 Hydrochemistry

Changes in the chemical composition of the Lake Żabińskie surface waters showed strong seasonality associated with spring blooms of phytoplankton and the processes of calcite precipitation as described by Bonk et al. (2015). This is illustrated by the variability of $Ca^{2+}$ concentrations and nutrient variability (Fig. 5). The highest concentrations of $Ca^{2+}$ were recorded during the winter season and the period of water mixing in early spring (> 70 mg $L^{-1}$). After the spring turnover, concentrations tended to drop as summer stratification developed (< 50 mg $L^{-1}$). Similarly, the highest mean concentrations of TP and TN in the epilimnion were measured in the winter and immediately after the ice-out (0.21 ± 0.71 mg $L^{-1}$ for TP, and 2.76 ± 2.78 mg $L^{-1}$ for TN). Also, maximum values are definitely highest during late winter and spring (Fig. 5). Afterward, during the summer stratification, epilimnetic values reached their annual minima as low as 0.05 mg $L^{-1}$ and 0.89 mg $L^{-1}$ for TP and TN, respectively. High nutrient concentrations in the surface waters during early spring resulted from nutrient accumulation in the hypolimnion during winter, which were transported to the epilimnion during lake spring turnover. Then, with the rapid air temperature rise and subsequent warming of the epilimnion, $CO_2$ solubility decreased, causing a quick increase of water pH. Intensive algae blooms led to $HCO_3^-$ depletion and $CaCO_3$ supersaturation. Finally, this led to rapid calcite precipitation, as seen in a drop of $Ca^{2+}$ concentrations and peaks in the deposition of carbonates accumulated in the sediment trap (Fig. 6).

## 4.3 Modern sedimentation

Sediment mass accumulation rates varied dramatically throughout the monitoring period (Fig. 6). The highest peaks in MAR were recorded each year shortly after the spring turnover (May/June), after which the values dropped towards the midsummer. In some years, another increase was visible near the fall turnover, e.g., in 2013, 2017, and 2021. The lowest values were observed during the winter with the lake under ice cover. The average MAR for all samples was 1.78 ± 1.82 g $m^{-2}$ $day^{-1}$ (n = 93). Maximum values of 9.02 g $m^{-2}$ $day^{-1}$ and 8.96 g $m^{-2}$ $day^{-1}$ were registered in 2020 and 2021, respectively. Overall, higher MAR values can be observed in years with intensive water column mixing, e.g., in 2016, 2020 and 2021.

TOC concentrations, reflecting deposition of organic matter, averaged 17.02 ± 5.79 %. The highest concentrations commonly occurred near the end of the summer stratification and during the fall turnover. With the onset of ice cover, concentrations usually decreased, and the lowest values occurred shortly after the spring mixing. Deposition of carbonates, represented by TIC, averaged 4.63 ± 2.18 %. In most years, TIC concentrations followed the pattern of MAR changes and formed a distinct peak shortly after the ice out, reaching up to 9.80 %, whereas the lowest values oscillating around 2.50 % were observed in the winter.

## 5 Data availability

The whole dataset from the period 2012–2021 contains monthly observations of the water column characteristics (water temperature, dissolved oxygen concentrations, pH, specific conductivity, and chlorophyll-a concentrations), water samples chemistry ($Ca^{2+}$, $Mg^{2+}$, $Na^+$, $K^+$, $SO_4^{2-}$, $Cl^-$, TP, TN), and sediment fluxes and their composition (MAR, TC, TOC, TIC, TN, TS). The dataset is available from the Bridge of Knowledge – Open Research Data Catalog from the following DOI: https://doi.org/10.34808/bsk4-eg58 (Tylmann et al., 2023). Daily meteorological data from the meteorological stations in

Kętrzyn and Mikołajki are publicly available and can be retrieved from the Institute of Meteorology and Water Management – National Research Institute open database (https://danepubliczne.imgw.pl/).

## 6 Conclusions

We present a decade-long and multi-parameter dataset with high temporal resolution for a typical eutrophic temperate lake system. Combined with available meteorological data, the dataset presented here can potentially be essential for modeling

physical and biogeochemical processes in lakes. By incorporating long-term monitoring data into models, we can improve our ability to make accurate predictions about future lake dynamics. This is particularly important due to ongoing climate change and human impact on lake ecosystems. Part of this dataset has been used to investigate the links between meteorological and limnological conditions and their influence on biochemical varve formation in Lake Żabińskie, showing a great potential in reconstructing paleoenvironmental conditions. However, there are still open questions related to

preservation of sub-seasonal meteorological events in the sediment records, which can be further translated to climatic signal over the longer time scales. Therefore, this unique dataset will be valuable for inter-site comparison of sediment fluxes variability and their relations to meteorological conditions, which may provide important regional or global context.

## Author contributions

Wojciech Tylmann prepared the manuscript with contributions from Alicja Bonk, Agnieszka Szczerba, and Maurycy

Żarczyński. All coauthors were involved in data collecting during field campaigns and laboratory work. Maurycy Żarczyński was responsible for raw data processing for the needs of this paper.

## Acknowledgements

This research was supported by project 2015/18/E/ST10/00325 funded by the National Science Centre and by project CLIMPOL (PSPB-086/2010) funded by the Polish-Swiss Research Programme. We thank our colleagues (Benjamin Amann,

Iwona Bubak, Christoph Butz, Artur Cysewski, Martin Grosjean, Iván Hernández-Almeida, Łukasz Pietruszyński, Tobias Schneider, Giulia Wienhues, Paul Zander) for sharing ideas and help with field and laboratory work.



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

**Table 1: Morphometric characteristics of Lake Żabińskie**

| | |
|---|---|
| Surface area (km$^2$) | 0.42 |
| Length of the shoreline (m) | 2846 |
| Shoreline development index | 1.2 |
| Max. length (m) | 1073 |
| Max. width (m) | 653 |
| Volume (thousand m$^3$) | 5072.8 |
| Max. depth (m) | 44.4 |
| Average depth (m) | 12.2 |
| Exposure index | 3.4 |







**Figure 1: Location and topography of the region (a) and Lake Żabińskie catchment (b), lake basin morphology with isobaths every 5 m (c), and sampling points ('O' stands for outflow and 'I' for inflowing streams). Digital Elevation Model courtesy of Polish Head Office of Geodesy and Cartography**






**Figure 2: Measurement setup (a), sediment trap (b), and data coverage (c). Abbreviations used in the panel (c) are explained in the sections 3.2, 3.3 and 3.4**




**Figure 3: Water temperature at different depths: (a) 1 m, (b) 10 m, (c) 20 m, (d) 30 m, (e) 40 m. The red line represents mean daily values while the grey bands show minimum and maximum values for each day. Blue bars indicate periods with ice cover on the lake.**




**Figure 4: Depth profiles of a) water temperature, b) oxygen concentration, c) pH, d) specific conductivity, and e) chlorophyll-a concentration through the water column of Lake Żabińskie**

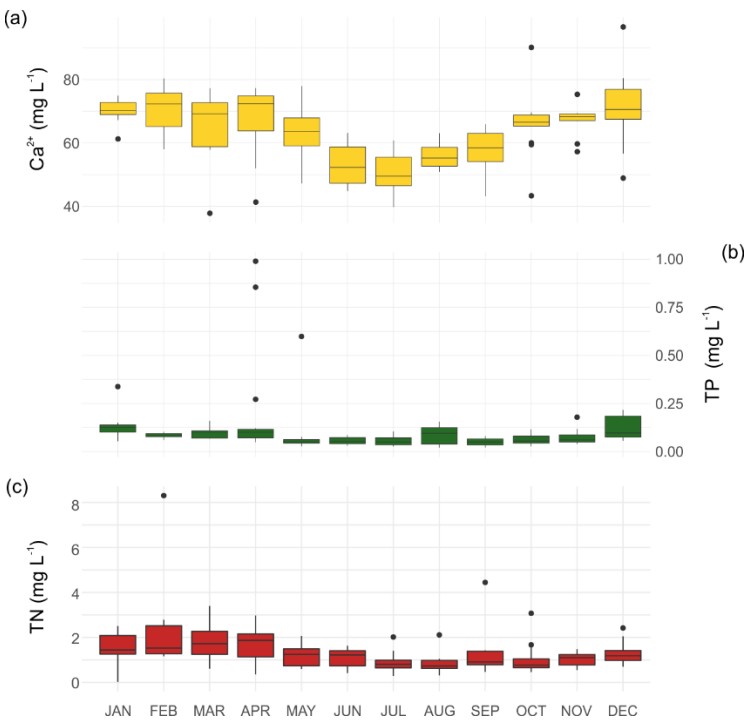

**Figure 5: Calcium (a), total phosphorus (b) and total nitrogen (c) concentrations in surface waters of Lake Żabińskie**

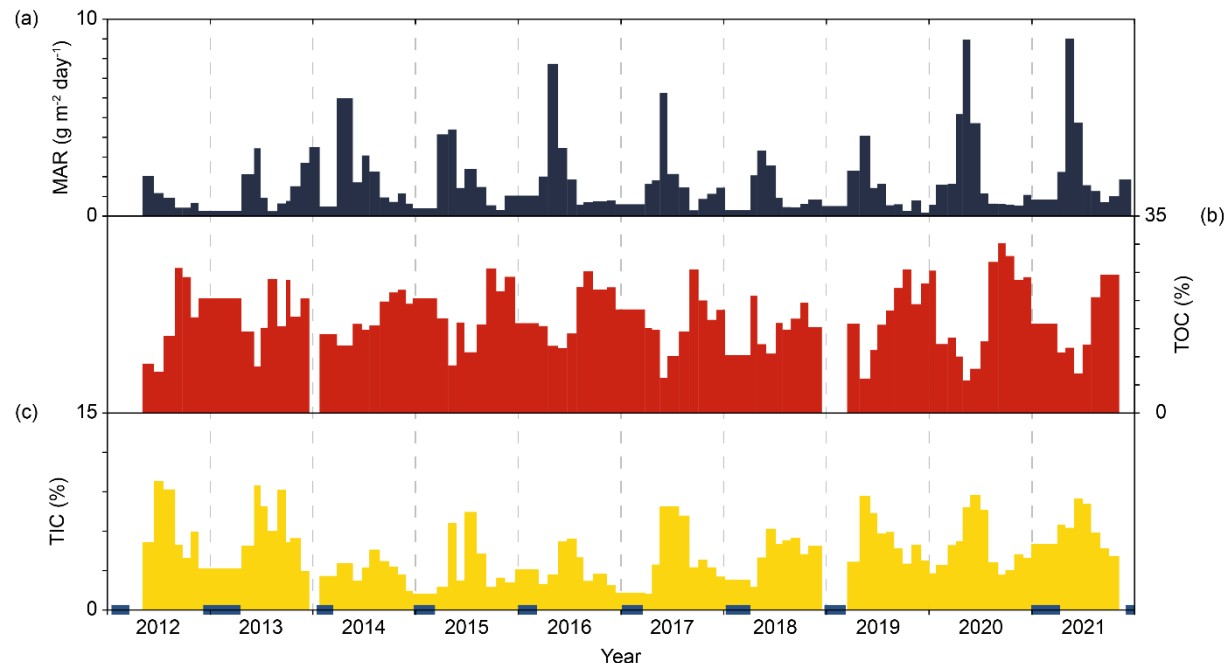

**Figure 6: Mass accumulation rate changes and weight % of major sediment components in the sediment trap samples**