# Peer review of "Investigating limnological processes and modern sedimentation at Lake Żabińskie, northeast Poland: a decade-long multi-variable dataset, 2012-2021"

_Earth System Science Data, 2023_

## Referee Comment (RC1)

Review of the paper entitled "Investigating limnological processes and modern sedimentation at Lake Żabińskie, northeast Poland: a decade-long multi-parameter dataset, 2012-2021" by Tylmann et al. submitted to ESSD.

This paper presents limnological data collected over a decade in a small lake of Northern Poland. While limnological data are not rare, this dataset is outstanding for two main reasons. First, it a decade-long continuous monitoring effort with a few short gaps, and this is exceptional. Second, the lake contains an exquisite sedimentary record made of pristine annual laminations, i.e., varves; this kind of record is rare. This sedimentary sequence has been studied in detail in many excellent scientific publications that presented paleoclimatic reconstructions and studies on the anthropogenic impact on lacustrine ecosystems. This dataset allows to decipher how climate influences limnological parameters, and hence the formation and the properties of varves.

I found the manuscript appropriate to support the publication of the dataset.
The dataset is unique as it required continuous funding for 10 years and 130 field campaigns. The dataset was useful to understand the formation of a varved sequence in Lake Żabińskie, a rare record of climate and environmental changes over the last 10 800 years. It can be used for modelling physical and biogeochemical processes in lakes, investigate the links between meteorological and limnological conditions, and hence preservation of sub-seasonal meteorological events in the sediment records, which can be further translated to climatic signal over the longer time scales.
The dataset is complete.

The data are accessible and well presented, easily understandable for most of them. I have not detected any faults.

The article is clear, well written and well presented with readable figures. English looks good, but I'm not a native speaker.

I have a few minor suggestions listed below.
Line 30: (e.g., one or two years, or even less).
Line 70: how can the catchment have elevation lower than the lake itself (reported being 117 amsl)?
Line 128: I suggest merging section 3.3 and 3.4 with a different name. The name for section 3.3 is a little misleading, as it could also apply to 3.4, 3.5 and 3.6.
Line 161 and following (Ice cover data): it is strange to give a single date to this parameter. Ice melting and freezing are processes occurring over several days. Maybe should you write something about the duration and/or accuracy (e.g., +/- n days) of this recorded parameter.
Figure 4: specific conductivity colour scheme is not very efficient to show something.
Figure 5: why plotting only the values for surface water when values for bottom waters are also available?

Metadata file (zab_metadata.pdf).

Hydrochemistry-tributaries: I suggest repeating what I1, I2 and I3 refer to (or a link to Figure 1).

Hydrochemistry: ions, I suggest spelling the elements out (na = sodium; k = potassium ….)

Sediment trap: explain why some data are missing with flags similar to the other datasets. Remind the depth of the trap in the metadata file.

Full resolution temperature data: the explanation of periods versus series is not clear. Would it be possible to explain this in a different way?

Finally, I tried to get the daily meteorological data from the meteorological stations from the Institute of Meteorology and Water Management – National Research Institute open database (https://danepubliczne.imgw.pl/), but the web site is in Polish only, and I have not been able to go further than the home page. Would it be possible to provide a more specific link to the two datasets of interest for this paper?

---

## Referee Comment (RC2)

The manuscript by Tylmann et al. provides a detailed description of datasets produced from a long-term environmental monitoring project at Lake Żabińskie in Poland. The lake contains annual laminations (varves) that have been analyzed in previous studies; the detailed monitoring presented here is useful for understanding the climatic signal recorded in these varves. This long-term (decade long) monitoring is particularly useful in the context of this varved record, and this dataset is highly worthy of publication in ESSD. The supplementary data are generally well organized and easy to navigate. My comments are minor and mainly focus on the organization of some sections of the manuscript and the framing of the datasets' significance:

1. Overall, I think the organization of sections 3 and 4 could be streamlined / redone a bit so they make more sense. The previous reviewer commented about combining 3.3 and 3.4, which could be a good start. But more broadly, I would make it very clear (and potentially group sections) based on how the data were collected (continuous instrumentation vs. discrete sampling) and/or the purpose of the data. Section 4 is organized into 1) water column data, 2) hydrochemistry data, and 3) modern sedimentation. Am I confused, or is the hydrochemistry data actually also water column data (that were collected from 40 m and 1 m, rather than every 10 m)? Perhaps Section 4 should be divided into a broader section of water related data, and another about sediment, with subsections as appropriate (i.e., for continuous vs discrete measurements, and/or for water properties vs hydrochemistry). Section 3 could follow the same general outline, so that it's easier for the reader to track the datasets presented as continuously measured vs sampled in the field (I found myself getting a little confused about this with the switching back and forth and different organization patterns in the different sections). The word "limnology" is used in the metadata to describe a subset of the water column data that are not hydrochemical; maybe this is a good framing to use in the paper organization, too.

2. In the introduction, the authors state that their results are relevant for modeling studies, and I think they're right. Though I realize this is not the main focus of the paper, I think the conclusion section would be strengthened with a brief discussion of the presented datasets and some more pointed recommendations for how they could be used in future limnological and modeling studies.

Specific/line-by-line comments:
L27-32: these statements follow a phrase about varves, specifically, and I think one of the main applications of these data is to understand what controls annual laminations, so it's a bit confusing that you then broaden out here to speak about lake sediments in general. Maybe rephrase or refine
L49-50: "lakes of temperate climate zones"
L51: "allow us to assess"
L95: how did these initial observations differ from "regular" ones?
L104-105: link to dataset
L122-123: I don't know what this means "daily mean values computed during the incremental database maintenance were recovered and used to fill the daily time series." What is the

"incremental database maintenance"? Do you mean during the regularly scheduled field sampling? Or something else? Either way, specify when / how often this occurred

Section 3.4: how often were water samples collected?

Section 3.6: more detail needed in the metadata; can you document how you acquired these data in each year presented, given the different methods outlined in this section? Can you give an estimate for the certainty/confidence somehow?

L175: "physiochemical"

Like the other reviewer, I did not find it immediately intuitive to use the links provided to access the meteorological data. Could this be made more seamless, and/or provided directly in the metadata?

Figures 3, 4, and 6: are these annual averages, or for a particular season?

Figure 3: this is Hobo data, correct?

Figure 4: can you specify which data were collected by continuous instrumentation vs. discrete measurements, and what we're seeing here?

Figure 5: what time period (in years) is this representing?

In the metadata, I think you need to specify somewhere what the IDs of the tributaries (O1, I1, etc.) correspond to). I realize this is somewhat done in figure 1, but these should have easily retrievable and identifiable coordinates that correspond to a title like "outflow 1" and then the abbreviation.

Also, in the "homogenized" temperature data, it's not clear to me what "series" and "period" refer to, and I don't think it's defined in the metadata (sorry if I missed that).

---

## Author Comment (AC1)

**Reviewer 1**

We would like to thank the Reviewer for his general very positive opinion about our manuscript as well as very constructive comments and suggestions. In the following we response to all of the comments (in bold italics).

Minor remarks:

***Line 30: (e.g., one or two years, or even less).***
Corrected.

***Line 70: how can the catchment have elevation lower than the lake itself (reported being 117 amsl)?***
Thank you for correcting our obvious mistake. The catchment has minimum elevation of 117 amsl which is the same as the lake level.

***Line 128: I suggest merging section 3.3 and 3.4 with a different name. The name for section 3.3 is a little misleading, as it could also apply to 3.4, 3.5 and 3.6.***
We changed the structure of sections 3 and 4 according to the suggestions of both reviewers. Now, section 3. Methods is divided into 3.1 Strategy for long-term monitoring, 3.2 Water column measurements and sampling (this section incorporates 3.2, 3.3 and 3.4 from the previous version), 3.3 Sediment trap sampling and analysis, and 3.4 Ice cover data. Section 4 Results is now divided into 4.1 Water column data time series and 4.2 Modern sedimentation.
In this way we simplified the structure and made sections 3 and 4 more comparable. Methods section 3.2 corresponds to results section 4.1 while section 3.3 corresponds to section 4.2. We do hope that the structure of new version is more logical and in line with the expectations of the reviewers.

***Line 161 and following (Ice cover data): it is strange to give a single date to this parameter. Ice melting and freezing are processes occurring over several days. Maybe should you write something about the duration and/or accuracy (e.g., +/- n days) of this recorded parameter.***
Thank you for this comment. Indeed, the process of ice cover formation and breakup take some time. It is very difficult to precisely indicate the beginning of the process without continuous (daily) monitoring of the lake surface. Also the end of the process is not easy to determine because remnants of ice cover can survive longer in specific locations, e.g. in the littoral zone overgrown with reeds. Since we rely mainly on observations of local citizens, the dates we present in the manuscript should be interpreted as follows: (i) the date of ice breakup is the first day with central part of the lake completely free of ice (but discontinuous ice cover still possible in the littoral zone), (ii) the date of ice cover formation is the first day with the whole central part of the lake covered with ice. We also used Landsat and Sentinel satellite imagery datasets to confirm the presence or absence of ice cover but due to cloud cover it was often not useful. We added an explanation to the metadata file as suggested by Reviewer 2 as well.

***Figure 4: specific conductivity colour scheme is not very efficient to show something.***
We modified the color scheme to be more expressive.

***Figure 5: why plotting only the values for surface water when values for bottom waters are also available?***

We decided to present in the figure surface water data because they illustrate strong seasonality and the process of calcite precipitation which is described in the text. The data for bottom waters are available in the data file and can be used for any specific reason. However, in the context presented in the manuscript we think it is not necessary to plot them.

***Metadata file (zab_metadata.pdf). Hydrochemistry-tributaries: I suggest repeating what I1, I2 and I3 refer to (or a link to Figure 1).***

We modified the metadata file. This reads now as:
- in_out (stream id): I* – inflow, O* – outflow; character:
  - I1: major inflow from Lake Purwin (54.13590731° N; 21.98398011° E)
  - I2: episodic inflow from the direct catchment (54.13160649° N; 21.98812808° E),
  - I3: major inflow from Żabinka village (54.12980987° N; 21.98246778° E),
  - O1: major outflow to Lake Gołdopiwo (54.13060358° N; 21.97381504° E).

***Metadata file (zab_metadata.pdf). Hydrochemistry: ions, I suggest spelling the elements out (na = sodium; k = potassium ….)***

We modified the metadata file. This part now reads:
- ions: mg L$^{-1}$; numeric:
  - na: sodium (Na$^+$),
  - k: potassium (K$^+$),
  - mg: magnesium (Mg$^{2+}$),
  - ca: calcium (Ca$^{2+}$),
  - cl: chloride (Cl$^-$),
  - so4: sulfate (SO$_2^{4-}$).

***Metadata file (zab_metadata.pdf). Sediment trap: explain why some data are missing with flags similar to the other datasets.***

We introduced the ND flag (no data). Lack of elemental data in two samples is related to the CNS analyzer failure (obviously wrong results, we were not able to repeat the measurements because of not enough sediment material left) while in one sample we measured TC/TN/TS but TOC/TIC could not be done due to lack of sediment material. Each of these cases is now marked with the ND flag.

***Metadata file (zab_metadata.pdf). Remind the depth of the trap in the metadata file.***

Added info to the metadata:
The active area of the trap was exposed at 2 m above the sediment surface and deployed at 2012.05.05.

***Metadata file (zab_metadata.pdf). Full resolution temperature data: the explanation of periods versus series is not clear. Would it be possible to explain this in a different way?***

We have rewritten this metadata information as follows:
"Series: one data series means that data logger operated continuously without any interruptions from the beginning of logging to the data offload.
Period: one period encompasses one or more series. Continued period means, that the end of one series is followed immediately by the beginning of the next series. Change from series to series means that only short logger offload time is introduced into the data. Continued

period allows uninterrupted data homogenization procedure. Change in the period variable means that the next series did not follow preceding one immediately and requires reset of the homogenization procedure."

***Finally, I tried to get the daily meteorological data from the meteorological stations from the Institute of Meteorology and Water Management – National Research Institute open database (https://danepubliczne.imgw.pl/), but the web site is in Polish only, and I have not been able to go further than the home page. Would it be possible to provide a more specific link to the two datasets of interest for this paper?***

Source of meteorological data is independent of our research and we cannot provide the files. However, in the section "Data availability" we provided wider explanation with a reference to a paper which explains how to retrieve data for specific meteorological station from the IMWM-NIR database. This part now reads:

„Long-term, daily meteorological data from the meteorological stations in Kętrzyn (approx. 40 km west of the lake; ID 12185) or in Mikołajki (approx. 50 km southwest of the lake; ID 12280) are publicly available from the Institute of Meteorology and Water Management – National Research Institute (https://danepubliczne.imgw.pl/). However, because of complicated manual data access procedure we suggest using the provided Application Programming Interface (API), for example using R "climate" package (Czernecki et al., 2020) and provided station IDs."

Czernecki, B.; Głogowski, A.; Nowosad, J. Climate: An R Package to Access Free In-Situ Meteorological and Hydrological Datasets for Environmental Assessment. Sustainability 2020, 12, 394. https://doi.org/10.3390/su12010394.

---

## Author Comment (AC2)

**Reviewer 2**

We would like to thank the Reviewer for the general positive opinion about our manuscript as well as very constructive comments and suggestions. In the following we response to all of the comments (in bold italics).

General comments:

1. ***Overall, I think the organization of sections 3 and 4 could be streamlined / redone a bit so they make more sense. The previous reviewer commented about combining 3.3 and 3.4, which could be a good start. But more broadly, I would make it very clear (and potentially group sections) based on how the data were collected (continuous instrumentation vs. discrete sampling) and/or the purpose of the data. Section 4 is organized into 1) water column data, 2) hydrochemistry data, and 3) modern sedimentation. Am I confused, or is the hydrochemistry data actually also water column data (that were collected from 40 m and 1 m, rather than every 10 m)? Perhaps Section4 should be divided into a broader section of water related data, and another about sediment, with subsections as appropriate (i.e., for continuous vs discrete measurements, and/or for water properties vs hydrochemistry). Section 3 could follow the same general outline, so that it's easier for the reader to track the datasets presented as continuously measured vs sampled in the field (I found myself getting a little confused about this with the switching back and forth and different organization patterns in the different sections). The word "limnology" is used in the metadata to describe a subset of the water column data that are not hydrochemical; maybe this is a good framing to use in the paper organization, too.***
We changed the structure of sections 3 and 4 according to the suggestions of both reviewers. Now, section 3. Methods is divided into 3.1 Strategy for long-term monitoring, 3.2 Water column measurements and sampling (this section incorporates 3.2, 3.3 and 3.4 from the previous version), 3.3 Sediment trap sampling and analysis, and 3.4 Ice cover data. Section 4 Results is now divided into 4.1 Water column data time series and 4.2 Modern sedimentation.
In this way we simplified the structure and made sections 3 and 4 more comparable. Methods section 3.2 corresponds to results section 4.1 while section 3.3 corresponds to section 4.2. We do hope that the structure of new version is more logical and in line with the expectations of the reviewers.

2. ***In the introduction, the authors state that their results are relevant for modeling studies, and I think they're right. Though I realize this is not the main focus of the paper, I think the conclusion section would be strengthened with a brief discussion of the presented datasets and some more pointed recommendations for how they could be used in future limnological and modeling studies.***
Thank you for this comment. We specified two potential directions of further use of our data: (i) modeling changes in lake water mixing regime and their ecological consequences, and (ii) investigations on the impact of climate warming on sedimentation processes in lakes. Now, this section reads as:

We present a decade-long and multi-parameter dataset with high temporal resolution for a typical eutrophic temperate lake system. Combined with available meteorological data, the dataset presented here can potentially be essential for modeling physical and

biogeochemical processes in lakes. By incorporating long-term monitoring data into models, we can improve our ability to make accurate predictions about future lake dynamics. This is particularly important due to ongoing climate change and human impact on lake ecosystems. Lake Żabińskie seems to be an extremely interesting object for studying the impact of increasing air temperature and decreasing seasonality (mild winters) on the lake water mixing regime. This, in turn, has a huge impact on the oxygen conditions in the lake along with changes in the nutrient cycling, and serious ecological consequences, e.g. massive blooms of harmful cyanobacteria species. Modeling these processes based on data from Lake Żabińskie may have a more universal application in relation to small and deep lakes of the European Lowland. Part of this dataset has already been used to investigate the links between meteorological and limnological conditions and their influence on biochemical varve formation in Lake Żabińskie, showing a great potential in reconstructing paleoenvironmental conditions. However, there are still open questions related to preservation of sub-seasonal meteorological events in the sediment records, which can be further translated to climatic signal over the longer time scales. Therefore, this unique dataset will be valuable for inter-site comparison of sediment fluxes variability and their relations to meteorological conditions, which may provide important regional or global context.

Specific comments:
***L27-32: these statements follow a phrase about varves, specifically, and I think one of the main applications of these data is to understand what controls annual laminations, so it's a bit confusing that you then broaden out here to speak about lake sediments in general. Maybe rephrase or refine.***
We specified that the whole paragraph is about varves by changing "lake sediments" with "varved lake sediments" as well as "sediment records" with "varved sediment records".

***L49-50: "lakes of temperate climate zones"***
Corrected.

***L51: "allow us to assess"***
This statement has more general meaning and is not related strictly to our work. Therefore, we prefer not to change this fragment.

***L95: how did these initial observations differ from "regular" ones?***
We performed first measurements during coring campaign in September 2011. However, we were not ready at that time to start the monitoring with monthly resolution and full range of parameters due to lack of equipment. Organization of field and lab infrastructure took some time, thus comparable data in terms of measurement techniques and range of parameters are available from 2012. Also, we are not sure about the quality of all results from 2011 because of technical issues. Therefore, we decided to present the measurements from 2012.

***L104-105: link to dataset***
Done.

***L122-123: I don't know what this means "daily mean values computed during the incremental database maintenance were recovered and used to fill the daily time series." What is the "incremental database maintenance"? Do you mean during the regularly***

***scheduled field sampling? Or something else? Either way, specify when / how often this occurred.***

We clarified and rewritten this part. This reads now as:

Due to hard drive failure and corruption of raw files with data from the logger installed at a depth of 1 m, no raw data is available from 2017.11.04 to 2019.01.30. However, we update our logger database and calculate daily means every time new data is acquired, which happens roughly once a year. We were able then to restore this lower-resolution data and use it to fill the gap.

***Section 3.4: how often were water samples collected?***

Water samples were collected with monthly or biweekly intervals at the same dates as limnological measurements. We added an explanation in the text.

***Section 3.6: more detail needed in the metadata; can you document how you acquired these data in each year presented, given the different methods outlined in this section? Can you give an estimate for the certainty/confidence somehow?***

Thank you for this comment. Indeed, the process of ice cover formation and breakup take some time. It is very difficult to precisely indicate the beginning of the process without continuous (daily) monitoring of the lake surface. Also the end of the process is not easy to determine because remnants of ice cover can survive longer in specific locations, e.g. in the littoral zone overgrown with reeds. Since we rely mainly on observations of local citizens, the dates we present in the manuscript should be interpreted as follows: (i) the date of ice breakup is the first day with central part of the lake completely free of ice (but discontinuous ice cover still possible in the littoral zone), (ii) the date of ice cover formation is the first day with the whole central part of the lake covered with ice. We also used Landsat and Sentinel satellite imagery datasets to confirm the presence or absence of ice cover but due to cloud cover it was often not useful. We added an explanation to the metadata file as suggested by Reviewer 1 as well.

***L175: "physiochemical"***

Corrected.

***Like the other reviewer, I did not find it immediately intuitive to use the links provided to access the meteorological data. Could this be made more seamless, and/or provided directly in the metadata?***

Source of meteorological data is independent of our research and we cannot provide the files. However, in the section "Data availability" we provided wider explanation with a reference to a paper which explains how to retrieve data for specific meteorological station from the IMWM-NIR database. This part now reads:

„Long-term, daily meteorological data from the meteorological stations in Kętrzyn (approx. 40 km west of the lake; ID 12185) or in Mikołajki (approx. 50 km southwest of the lake; ID 12280) are publicly available from the Institute of Meteorology and Water Management – National Research Institute (https://danepubliczne.imgw.pl/). However, because of complicated manual data access procedure we suggest using the provided Application Programming Interface (API), for example using R "climate" package (Czernecki et al., 2020) and provided station IDs."

Czernecki, B.; Głogowski, A.; Nowosad, J. Climate: An R Package to Access Free In-Situ Meteorological and Hydrological Datasets for Environmental Assessment. Sustainability 2020, 12, 394. https://doi.org/10.3390/su12010394.

***Figures 3, 4, and 6: are these annual averages, or for a particular season?***
Figure 3 - the red line represents mean daily values as explained in the figure caption.
Figure 4 – this figure presents the results from 130 measurement series (field campaigns in monthly or biweekly intervals). Each series consist of several parameters (water temperature, dissolved oxygen concentration, pH, specific conductivity, and chlorophyll-a concentration) in the water depth range of 0–40 m with 1 m intervals.
Figure 6 – this figure presents the daily fluxes, i.e., mass accumulation rates (MAR, g m−2 day−1) calculated for 93 samples collected from the sediment trap. Also values of TOC and TIC come form elemental analysis of the 93 sediment samples.

***Figure 3: this is Hobo data, correct?***
Yes, this dataset comes from measurements done with HOBO thermistors. We added this information to the figure caption.

***Figure 4: can you specify which data were collected by continuous instrumentation vs. discrete measurements, and what we're seeing here?***
This figure presents the results from 130 measurement series (field campaigns in monthly or biweekly intervals). Therefore, these are discrete measurements in contrast to Figure 3 where continuous thermistor datasets are presented.

***Figure 5: what time period (in years) is this representing?***
This figure presents data for the entire monitoring period (2012-2021). We added this information to the figure caption.

***In the metadata, I think you need to specify somewhere what the IDs of the tributaries (O1, I1, etc.) correspond to. I realize this is somewhat done in figure 1, but these should have easily retrievable and identifiable coordinates that correspond to a title like "outflow 1" and then the abbreviation.***
We modified the metadata file. This reads now as:
- in_out (stream id): I* – inflow, O* – outflow; character:
    - I1: major inflow from Lake Purwin (54.13590731° N; 21.98398011° E)
    - I2: episodic inflow from the direct catchment (54.13160649° N; 21.98812808° E),
    - I3: major inflow from Żabinka village (54.12980987° N; 21.98246778° E),
    - O1: major outflow to Lake Gołdopiwo (54.13060358° N; 21.97381504° E).

***Also, in the "homogenized" temperature data, it's not clear to me what "series" and "period" refer to, and I don't think it's defined in the metadata (sorry if I missed that).***
We have rewritten this metadata information as follows:
Series: one data series means that data logger operated continuously without any interruptions from the beginning of logging to the data offload.
Period: one period encompasses one or more series. Continued period means, that the end of one series is followed immediately by the beginning of the next series. Change from series to series means that only short logger offload time is introduced into the data. Continued period allows uninterrupted data homogenization procedure. Change in the period variable means that the next series did not follow preceding one immediately and requires reset of the homogenization procedure.

---

## Referee Report (RR1)

[referee-annotated manuscript omitted]

---

## Author Response (AR2)

**Reviewer 2**

We would like to thank the Reviewer for the general positive opinion about our revised manuscript as well as further constructive comments and suggestions. In the following we response to all of the comments (in bold italics).

1. *In response to my comments on the previous addition the authors provided the following information about figures 4 and 6:*
   *Figure 4 – this figure presents the results from 130 measurement series (field campaigns in monthly or biweekly intervals). Each series consist of several parameters (water temperature, dissolved oxygen concentration, pH, specific conductivity, and chlorophyll-a concentration) in the water depth range of 0–40 m with 1 m intervals.*
   *Figure 6 – this figure presents the daily fluxes, i.e., mass accumulation rates (MAR, g m−2 day−1) calculated for 93 samples collected from the sediment trap. Also values of TOC and TIC come from elemental analysis of the 93 sediment samples.*
   *This level of detail about the contents of the figure should be included in the captions.*

   We modified the figure captions accordingly.

2. *Generally the text reads quite well, but I found several minor grammatical errors that should be remedied by either the authors or a member of the journal's editorial team. Some examples from pages 2-3 of the ms:*
   *L 51: "allow to assess" should read "allow us to assess" or "allow for the assessment of" or something else grammatically correct*
   *L 67: "Lake Żabińskie is sited" should read "Lake Żabińskie is situated" or "Lake Żabińskie is located"*
   *L 69: "showing features of moraine landscape" should read "features of moraine landscapes" or "features typical of a moraine landscape" or similar.*

   Thank you for this comment. The whole text has been checked for English language by Reviewer 3. All suggested changes were accepted and introduced into the text.

3. *I am not sure whether or not it is an issue that the meteorological data described in the study are not provided. The authors address this in their comments on the previous version, explaining why they are not able to provide the data, but I'm not sure if this is acceptable given the journals mandate for FAIR data*

   We would like to emphasize once again that our paper presents data from our measurements – these are data on water temperature, chemistry, and sedimentation in Lake Żabińskie. Meteorological data do not come from our measurements and they are supplementary. Source of meteorological data is independent of our research and we cannot provide the files. However, in the section "Data availability" we provided wider explanation with a reference to a paper which explains how to retrieve data for specific meteorological station from the IMWM-NIR database.

**Reviewer 3**

We would like to thank the Reviewer for his general very positive opinion about our manuscript as well as very detailed checking of English language. We accepted all suggested changes.